# City Grade Classification Based on Connectivity Analysis by Luojia I Night-Time Light Images in Henan Province, China

**Zongze Zhao [1], Gang Cheng [1], Cheng Wang [2,\*], Shuangting Wang [1] and Hongtao Wang [1]**

[1] School of Surveying and Land Information Engineering, Henan Polytechnic University, Jiaozuo 454000, China; zhaozongze@hpu.edu.cn (Z.Z.); chenggang@hpu.edu.cn (G.C.); wst@hpu.edu.cn (S.W.); wht_31@hpu.edu.cn (H.W.)

[2] Key Laboratory of Digital Earth Science, Institute of Remote Sensing and Digital Earth, Chinese Academy of Sciences, Beijing 100094, China

\* Correspondence: wangcheng@radi.ac.cn; Tel.: +86-010-8217-8120

**Abstract:** City classification can provide important data and technical support for city planning and government decision-making. Traditional city classification mainly relies on the accumulation and analysis of census data, which requires a large time period and relies heavily on historical and statistical data. This paper mainly utilizes Luojia I Night-Time Light (NTL) images to analyze the rank classification of cities in Henan Province, China. Intensity values can be expressed as the mathematical surface of continuous human activities, and the basic characteristics of urban structures are determined by analogy with the topography of the earth. A connectivity analysis method for NTL images is proposed to analyze the connected regions of images at different intensity levels. By constructing a tree structure, different cities can be analyzed "crosswise" and "lengthwise" to generate a series of parametric information from connected regions of NTL images. Based on these parameters, 18 cities in Henan Province were classified and analyzed. The results show that these attribute information can be well used for city center detection and grade classification, and can meet the requirements of application analysis.

**Keywords:** grade classification; connectivity analysis; tree structure; NTL images; Henan Province

## 1. Introduction

In 2011, urbanization in China reached 51.27%, with the urban population surpassing the rural population for the first time. By the end of 2018, the urban population had reached 59.58% [1]. Many of China's largest cities, such as Shanghai, Beijing and Guangzhou, are pursuing multicenter, multicity spatial structures to disperse population and economic activity [2]. China has designed and implemented a city hierarchy system through urban master planning, urban regional planning and urban system planning to address transportation, housing and environmental problems associated with densely populated single-center cities [3].

"Central Place Theory" is a theory of urban geographical location, which was founded in 1933 by Walther Christaller [4]. It is considered that if the terrain is completely flat, the soil quality is the same, the population distribution is uniform, and the traffic convenience is equal, then the distribution of the town is uniform and regular in an equilateral hexagonal arrangement [5]. "Central Place Theory" has reference value to urban planning, production layout and regional planning. However, in fact, the terrain is not flat, the population density is not consistent, the consumption capacity and the traffic situation are not the same, so the distribution of the city is not regular, generally presents a hierarchical system, and is affected by the distribution of traffic trunk lines and resources [6].

The city hierarchy system is the relationship of the city hierarchy structure according to different levels of importance in the national or regional urban system [7]. The hierarchical structure of cities can reflect their development pattern, which will directly affect the economic development, tourism behavior, environmentally sustainable development and public expenditure of cities. Urban spatial structure refers to the spatial distribution and interaction between urban elements and various factors in the city, which includes the physical and perceptual environment [8–10]. Currently, there are several typical urban spatial structure models defined by Burgess [11]. The degree of urban spatial structure can be identified as concentration and dispersion [12], single center and multicenter [13], or clustered and dispersed [14], where density is considered to be the core concept of describing urban spatial structure [15]. Density is a measure of the concentration of human activity in a given unit of area. A large number of studies have shown that the density of urban built-up areas is closely related to the density of human activities [16]. On the one hand, physical density is the static property of spatial structure, which is related to the proportion of built-up areas in the urban environment. On the other hand, the density of social activities expresses the intensity of human interaction in the urban environment [17].

Thus, urban spatial structure can be measured by the spatial concentration of human activities. The density of human activity is affected by changing socioeconomic processes [17]. In recent years, the quantitative measurement of urban spatial structure [18] shows that urban spatial structure can be regarded as the collection of human interaction between different functional areas in urban areas, which refers to people's daily entertainment behavior [19]. Previous studies in urban center exploration and urban spatial structure analysis have mainly relied on census data and local knowledge [20,21]. These studies can be divided into three categories [21]: (1) identifying subcenters by setting thresholds for different census variables [22–24], (2) locating the local maximum value on the numerical surface of different census variables as the city center or subcenter [25–27] and (3) determining the urban center by regression residual analysis [28,29].

Although these methods have made great contributions to the empirical identification of urban subcenters and urban structures, they all have some recurring problems [21]. First, it is difficult to obtain the reference data of the appropriate region, which affects the final analysis results. Second, the previous method is only to detect and identify the city center, without analyzing the relationship between them and third, traditional methods can only be used for data analysis on a specific scale. To solve these problems, we adopted NTL (Night-Time Light) data instead of census data to conduct urban structure analysis, in which a connected operator method was used to divide NTL data and analyze its spatial relationship. The NTL data record the intensity, which reflects the scale of human activity. At present, many scholars use multi-temporal noctilucent images to analyze the urban expansion [30–33]. However, few scholars carry out detection and classification of urban centers. Chen et al. detected and identified the urban center with the attribute information such as intensity and area, but did not carry out a hierarchical analysis on it [21].

The early NTL composite data released by the National Oceanic and Atmospheric Administration's National Geophysical Data Center (NOAA/NGDC) were captured by the Defense Meteorological Satellite Program-Operational Linescan System (DMSP-OLS). DMSP-OLS composite data have been widely used in the delineation of cities [34,35], the estimation of socioeconomic indicators [36–38] and the spatiotemporal dynamic analysis of carbon dioxide emissions [39]. In early 2011, NOAA/NGDC released a new generation of NTL data from the day/night band (DNB) of the Visible Infrared Imaging Radiometer Suite (VIIRS) on the Suomi National Polar-orbiting Partnership (NPP) satellite. NPP-VIIRS NTL composite data also have a wider range of applications [40–43].

On 2 June 2018, Luojia I, developed by the Wuhan University team and relevant institutions, was successfully launched. Luojia I is currently the third satellite in the world with night light data shooting capability. The satellite makes up for the deficiency in night light data acquisition in China and has important historical significance and research value [44]. Compared with DMSP-OLS and NPP-VIIRS NTL composite data, the NTL of Luojia I has higher spatial resolution, which can better

show the detailed characteristics of the city. Luojia I data has been applied to the research of marine ship detection [45,46], construction land mapping [47], impervious surface detection [48], population mapping [49] and light pollution investigation [50]. However, due to the short orbit time of the Luojia I satellite, it is not possible to obtain the image data at long intervals, which makes it impossible to conduct regional change analysis.

This paper presents an urban center detection and classification method based on Luojia I NTL data. The remainder of it is organized as follows. The study area and data are presented in Section 2, followed by the description of methodology in Section 3. An analysis of research results is provided in Section 4. Finally, we offer a general discussion in Section 5 and a conclusion in Section 6.

## 2. Study Area and Data

### 2.1. Study Area

As shown in Figure 1, Henan Province is a provincial administrative region of China, the provincial capital of which, Zhengzhou, is located in central China. It is bounded by Anhui and Shandong provinces in the east, Hebei and Shanxi provinces in the north, Shaanxi Province in the west, and Hubei Province in the south. Henan has a total area of 167,000 square kilometers, and its population is more than 96 million (the third most populous province in China) [51]. Henan administers 17 provincial cities, including Zhengzhou, Kaifeng, Luoyang, Pingdingshan, Jiaozuo, Hebi, Xinxiang, Anyang, Puyang, Xuchang, Luohe, Sanmenxia, Nanyang, Shangqiu, Xinyang, Zhoukou and Zhumadian, and 1 provincial direct administration city, Jiyuan.

Henan, which has been regarded as "the center of the China" since ancient times, is located in the Jingjintang, Yangtze River delta, Pearl River delta and Chengdu-Chongqing urban belt [52]. It is also the hub of the national north-south and east-west communication arteries, the new Eurasian land bridge and the gateway to the six provinces in Northwest China. The unique and advantageous geographical location makes Henan a pivotal railway, highway, aviation, water conservancy, communications, energy and logistics hub. The hinterland of China is a national strategic comprehensive transport hub. The economy of the whole province is operating and improving steadily. According to preliminary calculations, the GDP (Gross Domestic Product) of the whole province reached 4.81 trillion yuan in 2018, an increase of 7.6%, one percentage point higher than the national average, and is fifth place in China and first place in the central and western provinces. The added value of industries above a designated size grew by 7.2% in 2018, one percentage point higher than the national average. The added value of the service sector reached 2,173,165 billion yuan in 2018, ranking first in the central region [53].

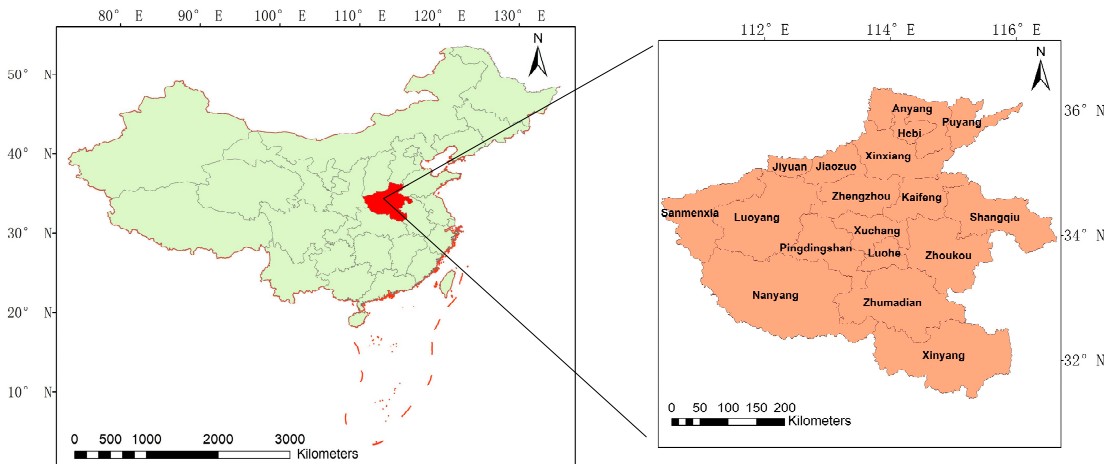

**Figure 1.** The area of study.

*2.2. Data*

The Luojia I NTL data is with a resolution of 130 m and a width of 250 km. Under usual conditions, global NTL remote sensing can be completed in 15 days. Since its launch in 2018, the Luojia I 01 star NTL data release system had registered more than 4000 users by the end of March 2019. It is mainly used to test and verify the domestic blank of "NTL remote sensing" technology and the national, urgently needed "low-orbit satellite navigation enhancement" technology in China. Equipped with a high-sensitivity luminous camera, the processing accuracy error can be controlled at approximately 100 m. The weighted euclide distance anisotropic diffusion noise suppression and the weak intersection plane region net adjustment method were used to improve the relative geometric accuracy of images to 1 pixel after orthogonalization correction, and the absolute geometric accuracy was better than 1.5 pixels.

As shown in Figure 2 and Table 1, there are 9 Luojia I NTL images covering the region of Henan province, most of which were acquired in October 2018, with resolution of 130 m. Among them, two of the images could not find images from October 2018, but only images from March 2019, the closest date. In order to facilitate processing, 9 images were spliced to obtain the whole image data of Henan province. The radiation intensity can be obtained by the following formula:

$$L = DN^{\frac{3}{2}} \times 10^{-10} \tag{1}$$

where, L is the radiation intensity after radiation correction, and DN is the gray value of the image.

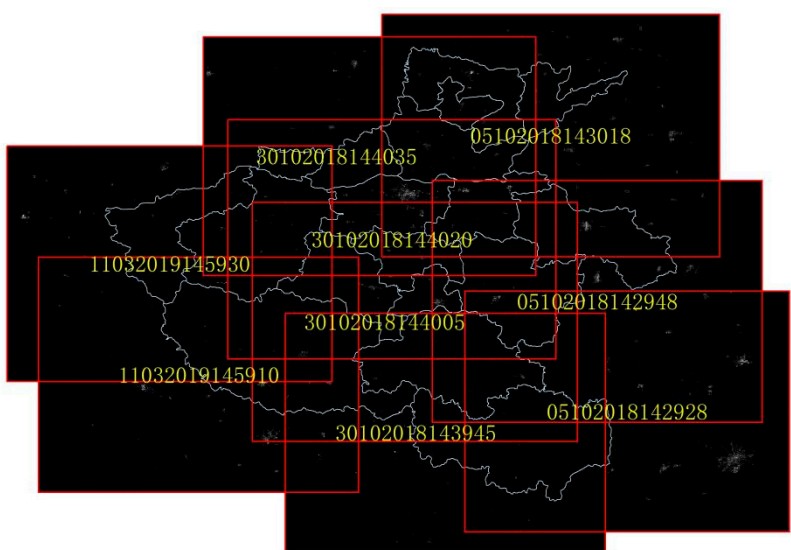

**Figure 2.** Luojia I NTL images distribution.

**Table 1.** List of Luojia I Night-Time Light (NTL) images data.

| Number | Date | Orbital Number | Resolution (m) |
|---|---|---|---|
| 1 | 05102018 | 142,928 | 130 |
| 2 | 05102018 | 142,948 | 130 |
| 3 | 05102018 | 143,018 | 130 |
| 4 | 30102018 | 143,945 | 130 |
| 5 | 30102018 | 144,005 | 130 |
| 6 | 30102018 | 144,020 | 130 |
| 7 | 30102018 | 144,035 | 130 |
| 8 | 11032019 | 145,910 | 130 |
| 9 | 11032019 | 145,930 | 130 |

## 3. Methodology

As shown in Figure 3, The method in this paper can be divided into four steps: (1) the connectivity analysis of NTL images at different levels, (2) construction of tree structure for the connected components at different levels, (3) attribute information acquisition of node in tree structure and (4) city center identification and classification. The detailed process is as shown in Figure 3.

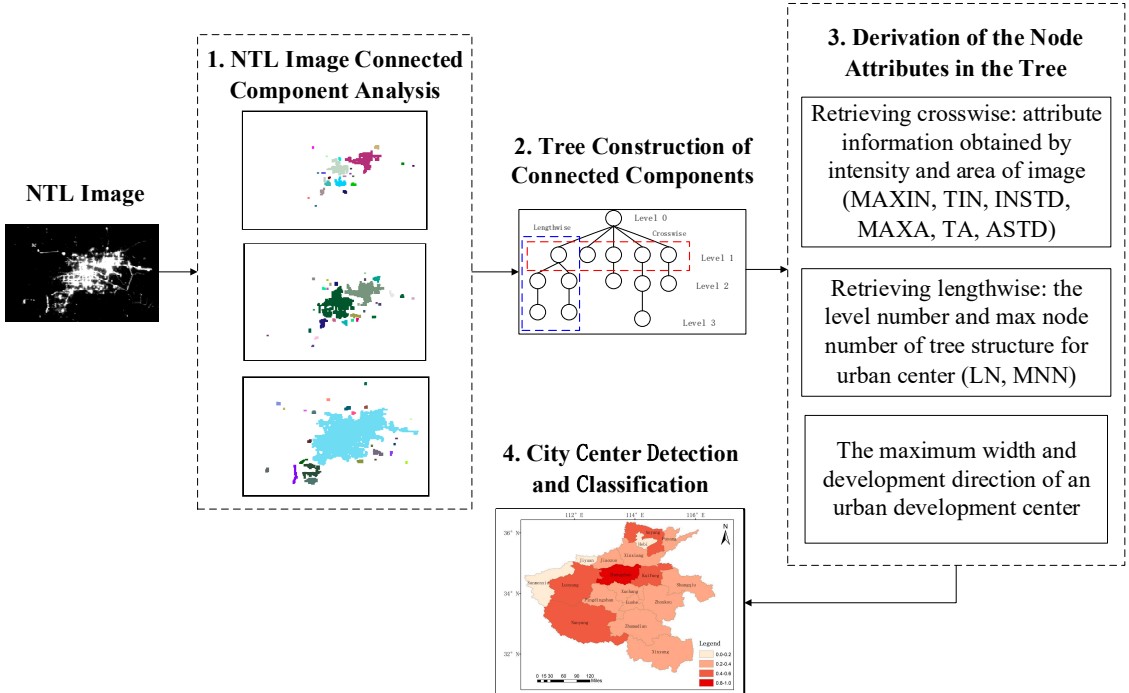

**Figure 3.** The technical flow chart of the paper.

### 3.1. Luojia 1 NTL Image Connected Component Analysis

Let H be a discrete Luojia 1 NTL image with E the domain of H, and T = $\{t_{min}, ... , t_i, ... , t_{max}\}$ is a finite set. A binary image B of H can be generated by thresholding H at level $t_i$, which is denoted by:

$$B_{t_i}(H) = \{h \in E | H(h) \geq t_i\} \tag{2}$$

and thresholding H by set T can achieve a stack of nested binary sets.

Given a connectivity class C, a subclass can be generated to reduce or increase members by modifying its associated connectivity opening. This is called second-generation connectivity [54] and its purpose is to model clusters or partitions of objects that cannot be otherwise captured. Clustering- or contraction-based second-generation connectivity are defined separately. In two cases, the second-generation connectivity analysis depends on the structural operator, such as opening or closing.

Clustering-based connectivity describes a set of image objects that can be thought of as a cluster of connected components if their relative distance is below a given threshold [55]. Contraction-based connectivity is a segmentation scheme in which a wide area of objects connected by a long and narrow structure in the original image can be regarded as separate objects. They are both determined by the size of the structuring element used along with an operator (opening or closing).

To partition connected components but not modify the existing edges, contraction-based connectivity is utilized to generate connected components on the different levels of the Luojia 1 NTL image. As shown in Figure 4, after the contraction-based connectivity analysis, the previously weakly connected regions (Figure 4a,c) can be separated well (Figure 4b,d).

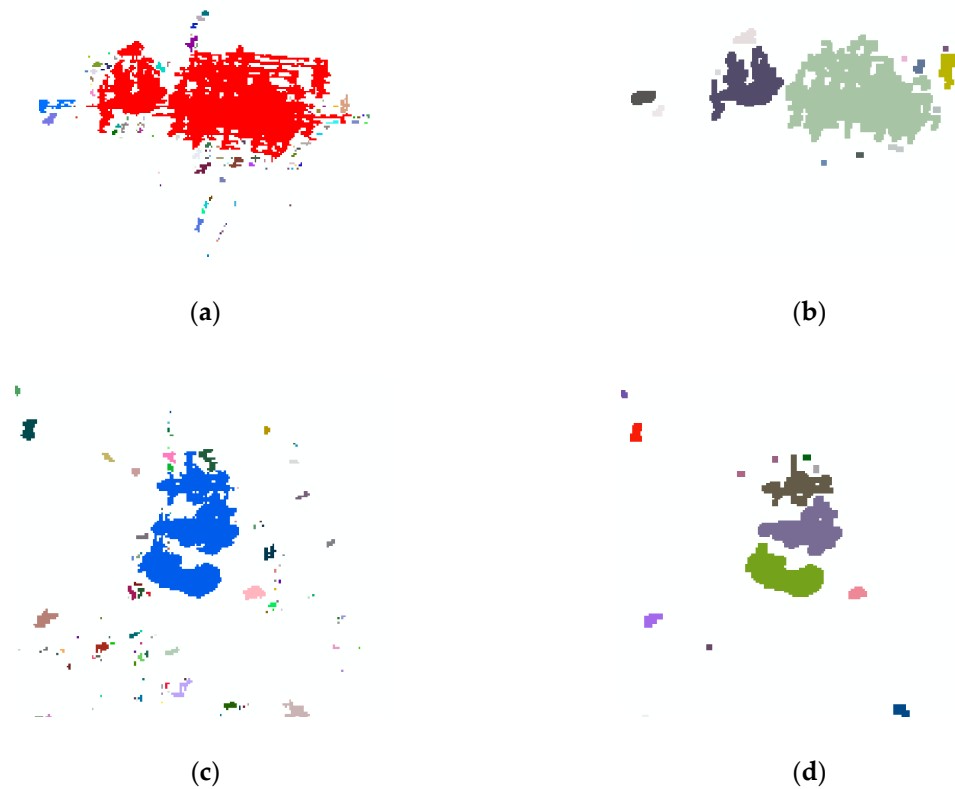

**Figure 4.** Contraction-based connectivity of two regions: (**a**) and (**c**) are the previously weakly connected regions, (**b**,**d**) are the regions after contraction-based connectivity analysis.

### 3.2. The Tree Construction of Connected Components

The purpose of this section is to construct a representation of the connected components represented by the tree hierarchical structure. The structure can provide more flexibility for the attribute expression of connected components at different levels. Usually, the max-tree/min-tree [56], the inclusion tree [57] and the binary partition tree (BPT) [58] are utilized to build the tree structure. One of the highest efficient tree representations is the max-tree. It constructs the connected components of the hierarchy set according to the inclusion relation of the hierarchy set. As shown in Figure 5, there are three levels from the NTL image, and the connected component in the red circle of the first level (Figure 5a) can be partitioned by connectivity analysis at the other two levels (Figure 5b,c), which can be expressed by one max-tree structure (Figure 5d). A node corresponds to a connected component at one level. The node at the minimum level in the NTL image is the entire image domain and is called the root of a tree. Nodes that do not link to another component at a higher level are called the leaf nodes of the tree. Therefore, the set of nodes and the links between them form the structure of the tree.

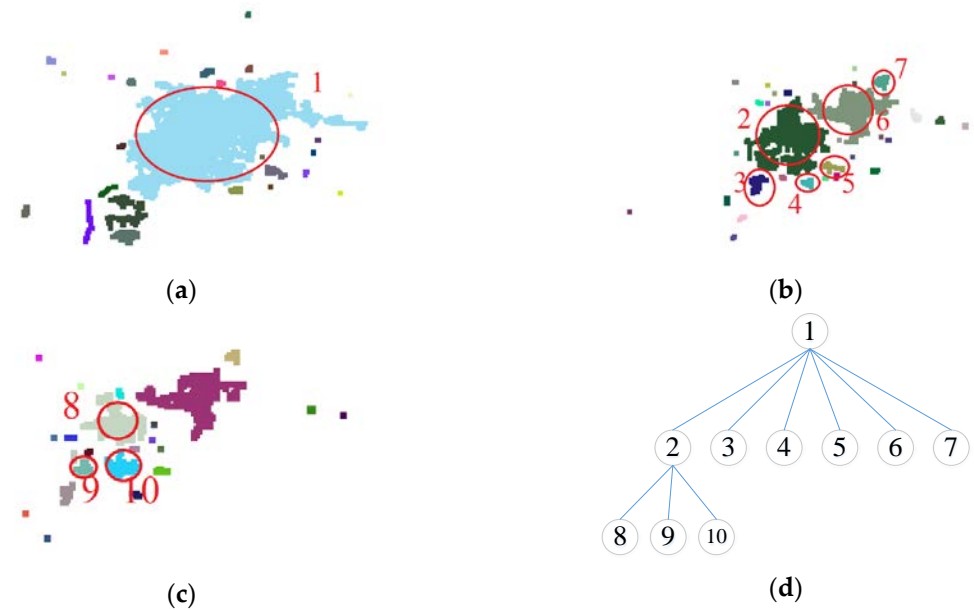

**Figure 5.** The tree construction of connected components: (**a**) the first layer of connectivity analysis results, (**b**) the second layer of connectivity, (**c**) the third layer of connectivity and (**d**) the constructed tree structure.

### 3.3. Derivation of the Node Attributes in the Tree

In the NTL max-tree, the node attributes are retrieved crosswise and lengthwise. As shown in Figure 6, the red and blue dashed rectangles represent "retrieving crosswise" and "retrieving lengthwise", respectively. The nodes at level 1 can present the main distribution of light areas, which can be analyzed by retrieving the attributes of nodes crosswise, and their tree structure can be assumed by retrieving nodes lengthwise.

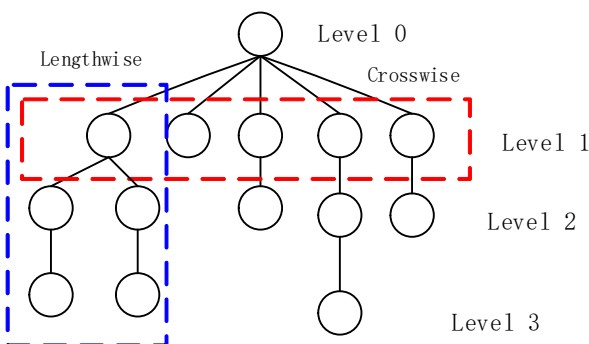

**Figure 6.** Tree structure analysis.

Usually, the intensity, area and shape information can be counted for the NTL image area. Therefore, for "retrieving crosswise", three statistics of NTL values were calculated at level 1 for each urban area, including maximum intensity (MAXIN), total intensity (TIN) and intensity standard deviation (INSTD) [21]. Similarly, basic urban area attributes were also computed to quantify the size characteristics of the urban area. The three attributes included maximum area (MAXA), total area (TA) and area standard deviation (ASTD). To analyze the width and direction of the urban development center area, the maximum width and direction (width and angle) of the development center area can be counted. The tree structure of each node at level 1 needs to be constructed, and the level number and max node number can be obtained from the tree structure to analyze the scale structure of urban areas. The level number (LN) and max node number (MNN) were calculated for the urban center in "retrieving lengthwise". The definitions of the above urban attributes are shown in Table 2.

**Table 2.** City analysis parameter list.

| Attribute | Definition |
|---|---|
| Maximum Intensity (MAXIN) | $\text{MAXIN} = \max_{i=1}^{N}\{r_i\}$<br>N is the number of nodes at level 1 for an urban area, and ri is the average night-time light intensity value of the ith node. |
| Total Intensity (TIN) | $\text{TIN} = \sum_{i=1}^{N} r_i$ |
| Intensity Standard Deviation (INSTD) | $\text{INSTD} = \sqrt{\frac{1}{N}\sum_{i=1}^{N}(r_i - \text{AIN})^2}$ |
| Maximum Area (MAXA) | $\text{MAXA} = \max_{i=1}^{N}\{a_i\}$<br>ai is the average area of the ith node. |
| Total Area (TA) | $\text{TA} = \sum_{i=1}^{N} a_i$ |
| Area Standard Deviation (ASTD) | $\text{ASTD} = \sqrt{\frac{1}{N}\sum_{i=1}^{N}(a_i - \text{AVA})^2}$ |
| Width | The maximum width of an urban development center. |
| Angle | The development direction of the urban development center. |
| Level Number (LN) | LN is the level number of a tree structure for urban center. |
| Max Node Number (MNN) | MNN is the max node number of a tree for urban center. |

### 3.4. City Center Detection and Classification

The urban center is defined as a continuous region where the employment density and total employment are both higher than their preset critical values. This means that an important and meaningful urban center should have a sustained, significantly higher NTL intensity and larger area than its surroundings [21]. For a complex urbanized area, many tree structures can be constructed. The "primary tree" is the largest NTL or area tree structure in the urban area, covering the urban area with the highest NTL intensity and largest area and the area with the most concentrated human activities.

Urban area is denoted by a tree structure with a single or multicenter structure. For a single central urban structure, the tree structure has only one branch. Only one leaf node is left at the last level of the tree, and the connected component corresponding to this leaf node represents the spatial range of the urban center. The tree structure of multicenter urban areas usually has two or more branches, which have multiple leaf nodes representing the basic urban center. The level number of the tree structure reflects the complexity of the multicenter urban structure. Therefore, the intensity, areal extent and tree structure of connected components at level 1 can determine and reflect the level of urban development.

The parameters LN and MMN of the tree structure can reflect the width and length of the urban center. We mainly combine the TIN, TA, LN and MMN indexes to classify cities in Henan Province. The following formula is used to calculate the comprehensive indicator, Q, of each city:

$$Q_i = a\frac{\text{TIN}_i}{\text{TIN}_{max}} + b\frac{\text{TA}_i}{\text{TA}_{max}} + c\frac{\text{LN}_i}{\text{LN}_{max}} + d\frac{\text{MMN}_i}{\text{MMN}_{max}} \tag{3}$$

where a, b, c and d are the weight coefficients of the four indexes. The value of the weight can be set to different large sizes according to the degree of emphasis of a certain index, but the sum of the weights must be equal to 1. Here, all four weights are set to 0.25, and they are equally important. Parameters $\text{TIN}_i$, $\text{TA}_i$, $\text{LN}_i$ and $\text{MMN}_i$ are the four indicators of one city. Parameters $\text{TIN}_{max}$, $\text{TA}_{max}$, $\text{LN}_{max}$ and $\text{MMN}_{max}$ are the largest indicators of any city.

## 4. Results

### 4.1. City Center Detection and Classification

The distribution map of Henan Province can be obtained according to the existing vector data, by which NTL image data can be divided into different city units. Through the NTL image data,

the urban spatial hierarchy is analyzed for each city "crosswise" and "lengthwise". Statistic values are calculated at level 1 for each urban in "crosswise", and the distribution of the connected components is shown in Figure 7. As shown in Table 3, the connected component attribute information of 18 cities is counted at level 1, and the maximum values per column are shown in bold font. From the results in Table 3, we can see that Nanyang city has the largest intensity value (4.45), which is related to the population size (10.01 million). As shown in Figure 8, Zhengzhou and Nanyang have larger populations [59], which also have higher intensity values. This also indicates that the NTL image intensity information can reflect the population information. Except for Zhengzhou and Nanyang, the intensity values of most cities are approximately 1, and the intensity values of Hebi, Jiyuan and Luohe were below 0.5.

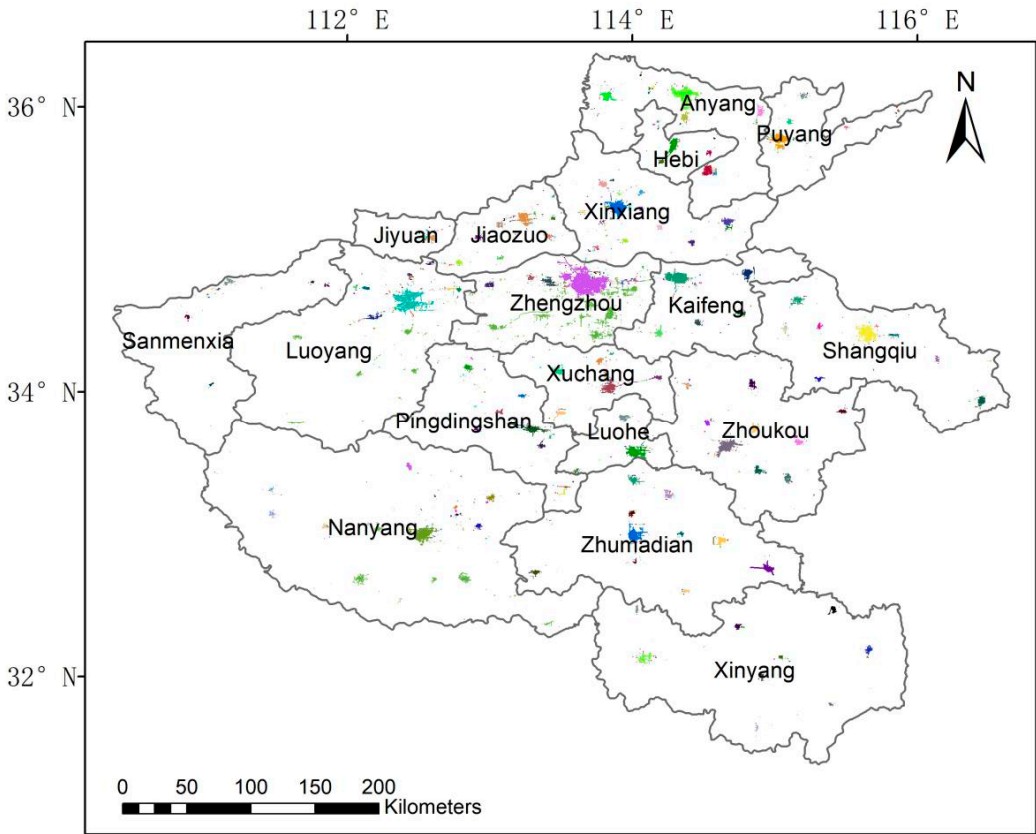

**Figure 7.** The distribution of connected components at level 1.

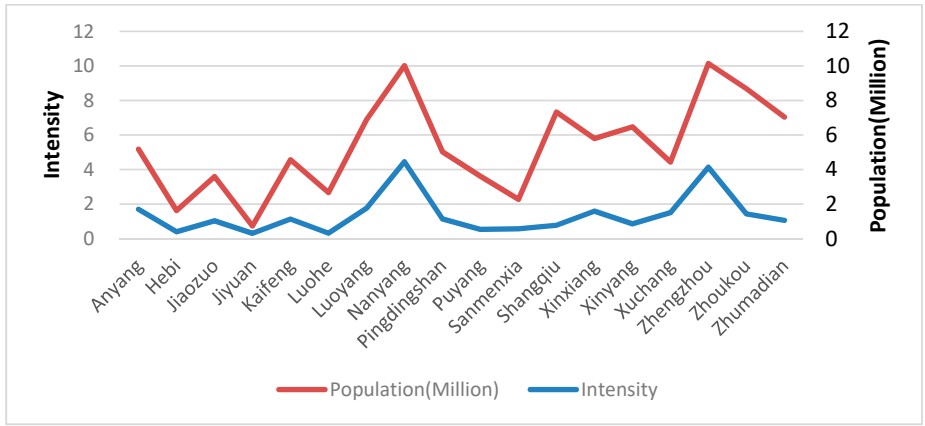

**Figure 8.** The relationship between population and intensity.

**Table 3.** City parameters of Henan Province, and the maximum values per column are shown in bold font.

| City | MAXIN | TIN | INSTD | MAXA (km2) | TA (km2) | ASTD | Width (km) | Angle | LN | MNN |
|---|---|---|---|---|---|---|---|---|---|---|
| Anyang | 0.34 | 1.70 | 0.05 | 158.88 | 327.45 | 959.05 | 22.88 | 0.23 | 16 | 7 |
| Hebi | 0.10 | 0.40 | 0.03 | 58.10 | 89.86 | 603.28 | 17.42 | −1.12 | 5 | 3 |
| Jiaozuo | 0.18 | 1.03 | 0.03 | 67.38 | 153.72 | 400.80 | 12.87 | −0.14 | 5 | 3 |
| Jiyuan | 0.17 | 0.31 | 0.03 | 21.68 | 27.14 | 219.96 | 6.76 | 0.08 | 2 | 2 |
| Kaifeng | 0.25 | 1.13 | 0.04 | 132.26 | 262.95 | 864.91 | 22.23 | 0.01 | 15 | 3 |
| Luohe | 0.17 | 0.31 | 0.03 | 101.38 | 138.66 | 1019.44 | 19.76 | 0.11 | 7 | 4 |
| Luoyang | 0.20 | 1.76 | 0.03 | 235.64 | 394.87 | 1042.34 | 34.84 | 0.36 | 7 | 8 |
| Nanyang | **0.95** | **4.45** | **0.11** | 138.60 | 325.66 | 623.66 | 24.05 | 0.05 | 7 | 4 |
| Pingdingshan | 0.15 | 1.14 | 0.03 | 85.90 | 182.57 | 509.41 | 34.45 | 0.14 | 6 | 5 |
| Puyang | 0.12 | 0.53 | 0.02 | 95.40 | 128.73 | 667.98 | 17.29 | −0.05 | 7 | 4 |
| Sanmenxia | 0.12 | 0.57 | 0.03 | 21.04 | 48.69 | 192.61 | 8.84 | −0.09 | 2 | 3 |
| Shangqiu | 0.14 | 0.78 | 0.02 | 132.43 | 261.81 | 809.25 | 17.42 | 0.22 | 15 | 2 |
| Xinxiang | 0.16 | 1.59 | 0.03 | 107.08 | 276.70 | 605.76 | 17.55 | 0.10 | 5 | 4 |
| Xinyang | 0.14 | 0.86 | 0.03 | 39.41 | 141.96 | 351.44 | 12.09 | −0.30 | 3 | 3 |
| Xuchang | 0.44 | 1.51 | 0.05 | 75.85 | 189.21 | 516.12 | 21.32 | −0.33 | 6 | 4 |
| Zhengzhou | 0.18 | 4.14 | 0.03 | **461.12** | **907.06** | **1433.26** | **42.51** | 0.35 | **19** | **10** |
| Zhoukou | 0.40 | 1.44 | 0.04 | 108.75 | 287.86 | 669.91 | 22.23 | −0.36 | 14 | 2 |
| Zhumadian | 0.17 | 1.06 | 0.03 | 91.19 | 301.50 | 706.60 | 18.59 | **1.32** | 7 | 3 |

As shown in Table 3, for the standard deviation of the NTL image intensity value, only the standard deviation of Nanyang city is greater than 0.1, which indicates that the development of Nanyang city is uneven among different counties. The standard deviation of Anyang and Xuchang is greater than 0.05 and less than 0.1. The rest of the cities have standard deviations between 0.02 and 0.04, which shows that the development of most cities, except Nanyang city, is relatively balanced. Although Zhengzhou has a large intensity value, the standard deviation is only 0.03, indicating that all the subordinate counties of Zhengzhou have good development.

As seen from the area information in Table 3, Zhengzhou has a relatively large area, which can reach more than 900 km². In addition to Zhengzhou, Luoyang, Nanyang, Anyang and Zhumadian cities have areas of more than 300 km². Kaifeng, Shangqiu, Xinxiang and Zhoukou cities have areas between 200 and 300 km², and the other cities are approximately 100 km². Only Jiyuan and Sanmenxia are under 50 km². For the standard deviation of the area, only Luohe, Luoyang and Zhengzhou have standard deviations above 1000, while the standard deviations of the other cities are below 1000. In addition, although Nanyang city has a large population, the area is not particularly large, which indicates that Nanyang city has a large population density.

Each city has its own development center, which determines the degree of development in the city. Because the time since the launch of the Luojia I satellite in June 2018 is not long and there are no data of long time series, the urban expansion direction cannot be analyzed through multiperiod images but can only be analyzed through spatial structure. As shown in Figure 9, this paper utilizes "width" and "angle" to describe the development trend of cities in Henan Province. They represent the length and angle of the longest span of a city. As shown in Table 3, Zhengzhou, the provincial capital, has the largest span of 42.51 km, and its development direction is mainly northwest and southeast. The maximum spans of Luoyang and Pingdingshan cities are between 30 and 40 km, and they also developed in the northwest and southeast directions. The maximum spans of Anyang, Kaifeng, Nanyang, Xuchang and Zhoukou cities are between 20 and 30 km. The development direction of Anyang city is northwest and southeast, the directions of Kaifeng and Nanyang cities are close to east-west and the directions of Xuchang and Zhoukou cities are southwest and northeast. The maximum spans of Hebi, Jiaozuo, Luohe, Puyang, Shangqiu, Xinxiang, Xinyang and Zhumadian cities are between 10 and 20 km. The development directions of Luohe, Shangqiu and Zhumadian cities are northwest and southeast. However, the directions of Luohe and Shangqiu cities are closer to east-west, while Zhumadian city is closer to north-south, the directions of Puyang and Xinxiang cities are close to east-west and the

directions of Hebi, Jiaozuo and Xinyang cities are southwest and northeast. The maximum spans of Jiyuan and Sanmenxia cities are under 10 km, and their development directions are close to east-west.

The tree structure of the central city can be listed for the "lengthwise" analysis. The tree structures of the urban development center in Pingdingshan are shown in Figure 10a. Each node of the tree structure represents a connected region with an area greater than 1.44 km$^2$ (100 pixels) on each level of the NTL image. The LN and MNN of Pingdingshan city are 6 and 5, respectively. The distribution of connected regions at each level is shown in the Figure 10b–g. Among them, the number of connected regions at level 5 reaches the maximum of 5. The LN and MNN indicators for other cities are shown in the last two columns of Table 3. In addition to Zhengzhou city, Anyang, Kaifeng, Shangqiu and Zhoukou cities have LN indexes over 10. Apart from Zhengzhou, only Anyang and Luoyang cities have MNN indexes above 5. According to the LN and MNN indicators, all cities can be divided into two development modes: multi-edge and multicenter. The multi-edge development mode has a multilevel structure and few nodes in the urban center tree (e.g., Kaifeng and Shangqiu cities), while the multicenter development mode has a structure with few levels and many nodes (e.g., Luoyang city).

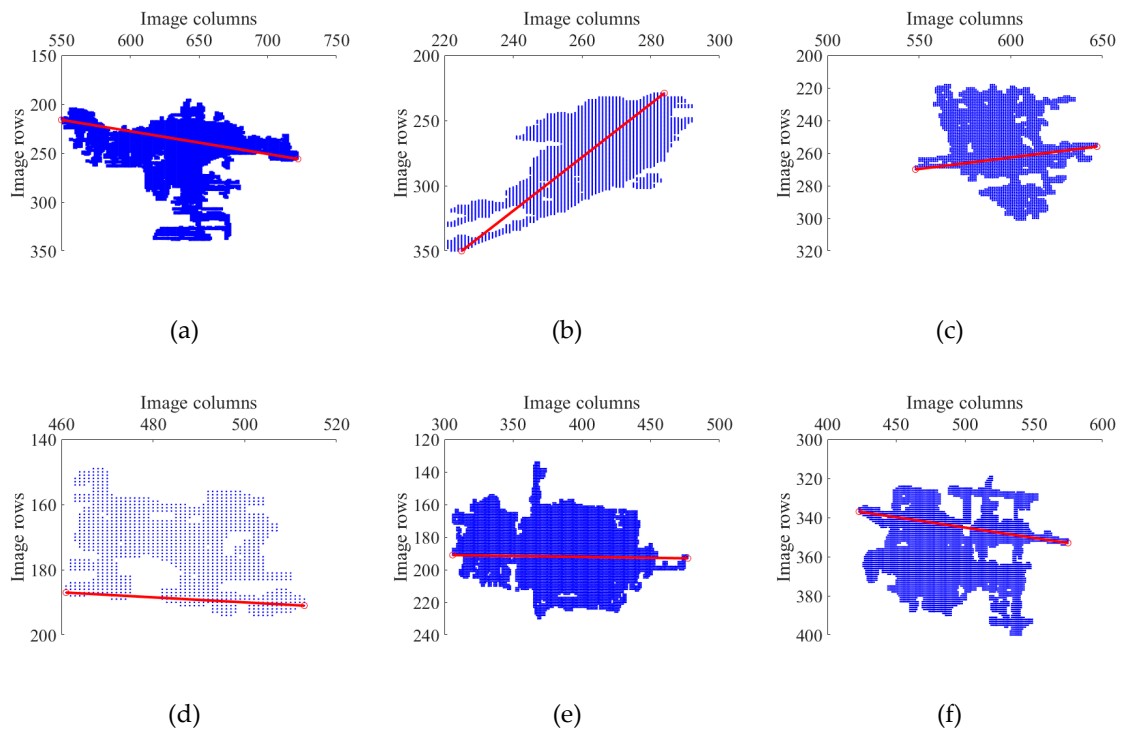

**Figure 9.** *Cont.*

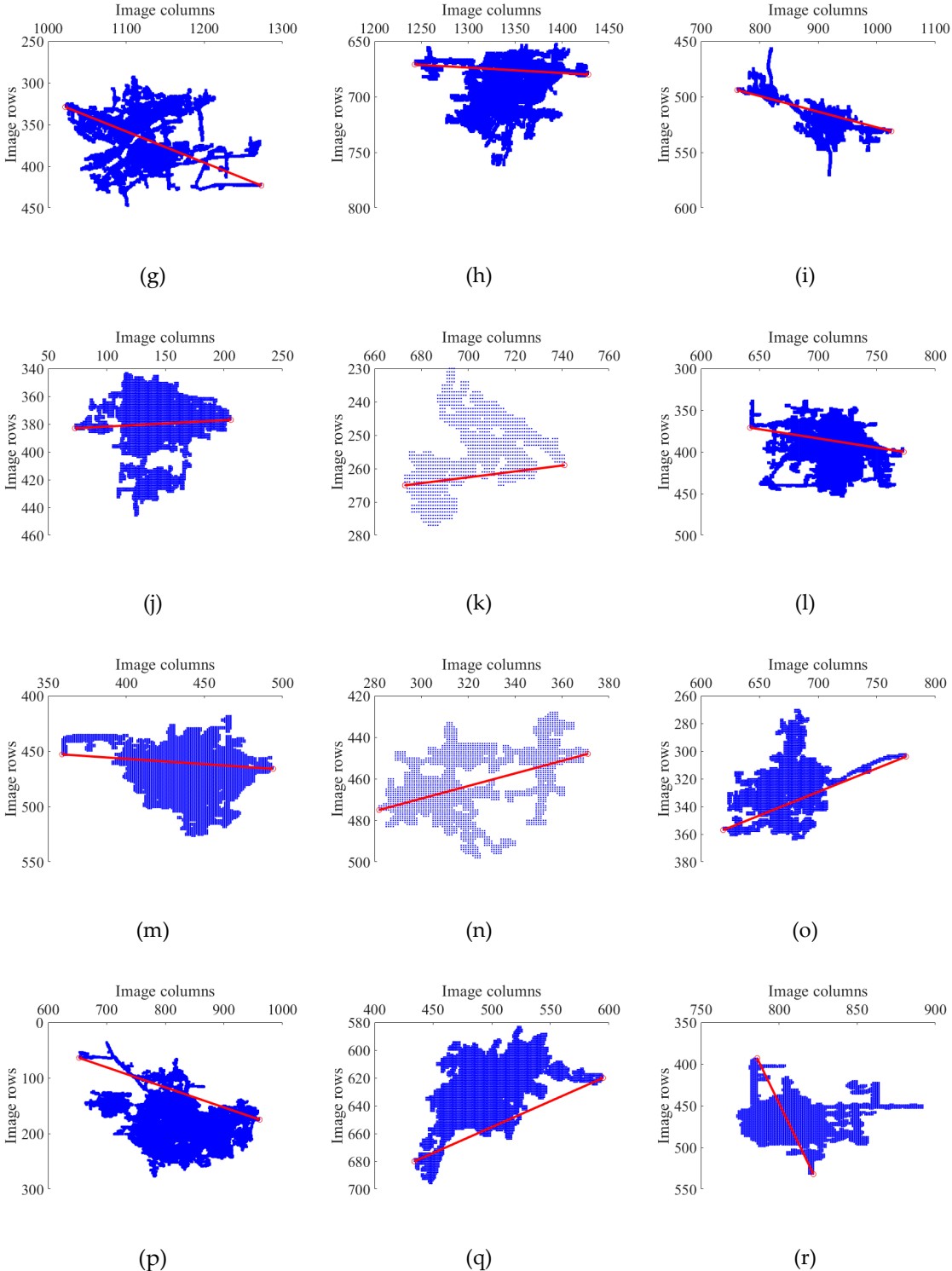

**Figure 9.** The "width" and "angle" of cities in Henan Province: (**a**)Anyang, (**b**) Hebi, (**c**) Jiaozuo, (**d**) Jiyuan, (**e**) Kaifeng, (**f**) Luohe, (**g**) Luoyang, (**h**) Nanyang, (**i**) Pingdingshan, (**j**) Puyang, (**k**) Sanmenxia, (**l**) Shangqiu, (**m**) Xinxiang, (**n**) Xinyang, (**o**) Xuchang, (**p**) Zhengzhou, (**q**) Zhoukou, (**r**) Zhumadian.

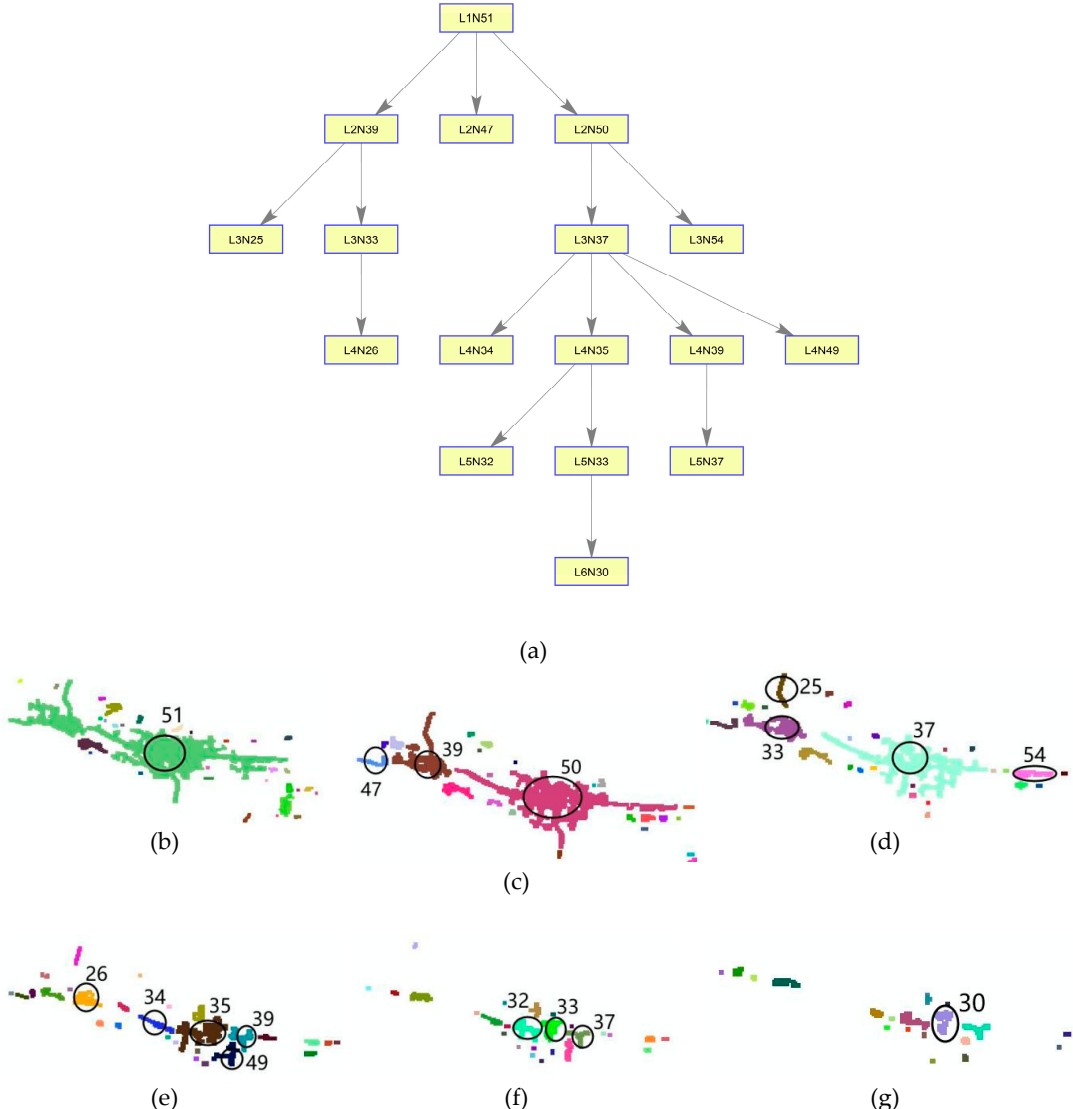

**Figure 10.** The central city tree structure of Pingdingshan: (**a**) the tree structure, (**b**) the distribution of connected regions corresponding to nodes at level 1, (**c**) the distribution of connected regions corresponding to nodes at level 2, (**d**) the distribution of connected regions corresponding to nodes at level 3, (**e**) the distribution of connected regions corresponding to nodes at level 4, (**f**) the distribution of connected regions corresponding to nodes at level 5 and (**g**) the distribution of connected regions corresponding to nodes at level 6.

### 4.2. City Classification of Henan Province

The key to the comparison of urban comprehensive competitiveness is the comparison of the agglomeration and diffusion functions of the urban economy. The intensity of the NTL image can reflect the degree of urban population aggregation and indirectly reflect the degree of economic aggregation, while the area can reflect the dispersive degree of the economy. Therefore, indicators TIN and TA can be used to reflect the comprehensive capacity of the city. The comprehensive index, P, can be calculated by the following formula:

$$P_i = a\frac{TIN_i}{TIN_{max}} + b\frac{TA_i}{TA_{max}} \tag{4}$$

where a and b are the weight coefficients of the two indexes. In this article, both a and b are set to 0.5. Parameters $TIN_i$ and $TA_i$ are the two indicators of one city, and $TIN_{max}$ and $TA_{max}$ are the largest indicators of any city.

To verify the correlation between this index and the economy, the GDP ranking of 18 cities in Henan Province was introduced for comparative analysis. The comparison results are shown in Table 4. The index can be used to rank 18 cities in Henan Province and compare them with the GDP ranking of these cities to determine their differences. The proportion of these differences to the maximum difference can be calculated. It can be seen from the statistical results that the maximum difference is 4, accounting for 0.24. The average weight of the 18 cities was just 0.11. It can be seen from this that this index and the GDP index have very close similarity and can represent the economic development indicator as a whole.

**Table 4.** The indicators in this paper are compared with GDP (Gross Domestic Product) indicators.

| City | Index | Ranking | GDP Ranking | Difference | Proportion |
|------|-------|---------|-------------|------------|------------|
| Anyang | 0.37 | 4 | 7 | 3 | 0.18 |
| Hebi | 0.09 | 16 | 17 | 1 | 0.06 |
| Jiaozuo | 0.20 | 12 | 10 | 2 | 0.12 |
| Jiyuan | 0.05 | 18 | 18 | 0 | 0.00 |
| Kaifeng | 0.27 | 9 | 13 | 4 | 0.24 |
| Luohe | 0.11 | 15 | 16 | 1 | 0.06 |
| Luoyang | 0.42 | 3 | 2 | 1 | 0.06 |
| Nanyang | 0.68 | 2 | 3 | 1 | 0.06 |
| Pingdingshan | 0.23 | 11 | 12 | 1 | 0.06 |
| Puyang | 0.13 | 14 | 14 | 0 | 0.00 |
| Sanmenxia | 0.09 | 17 | 15 | 2 | 0.12 |
| Shangqiu | 0.23 | 10 | 8 | 2 | 0.12 |
| Xinxiang | 0.33 | 5 | 8 | 3 | 0.18 |
| Xinyang | 0.17 | 13 | 9 | 4 | 0.24 |
| Xuchang | 0.27 | 8 | 4 | 4 | 0.24 |
| Zhengzhou | 0.96 | 1 | 1 | 0 | 0.00 |
| Zhoukou | 0.32 | 6 | 5 | 1 | 0.06 |
| Zhumadian | 0.29 | 7 | 11 | 4 | 0.24 |

As shown in Figure 11, cities in Henan Province are evenly divided into five grades according to comprehensive indicators, Q, calculated by Formula (3). Zhengzhou is far ahead of other cities in the first tier (0.8–1.0). There are no cities in the second tier (0.6–0.8), and the third tier (0.4–0.6) includes Anyang, Kaifeng, Luoyang and Nanyang cities. Jizaozuo, Luohe, Pingdingshan, Puyang, Shangqiu, Xinxiang, Xinyang, Xuchang, Zhoukou and Zhumadian cities are in the fourth tier (0.2–0.4), and Hebi, Sanmenxia and Jiyuan cities are in the fifth tier (0.0–0.2). The values of the comprehensive indicators, Q, for each city are shown in Table 5. In addition, compared with the 2017 and 2018 results of China's national classification, the results of Anyang and Kaifeng cities were different from those in 2017 or 2018. National classification is based on economic, cultural, educational, development, industry and other indicators. Cultural education cannot be fully reflected in the comprehensive indicators in this paper.

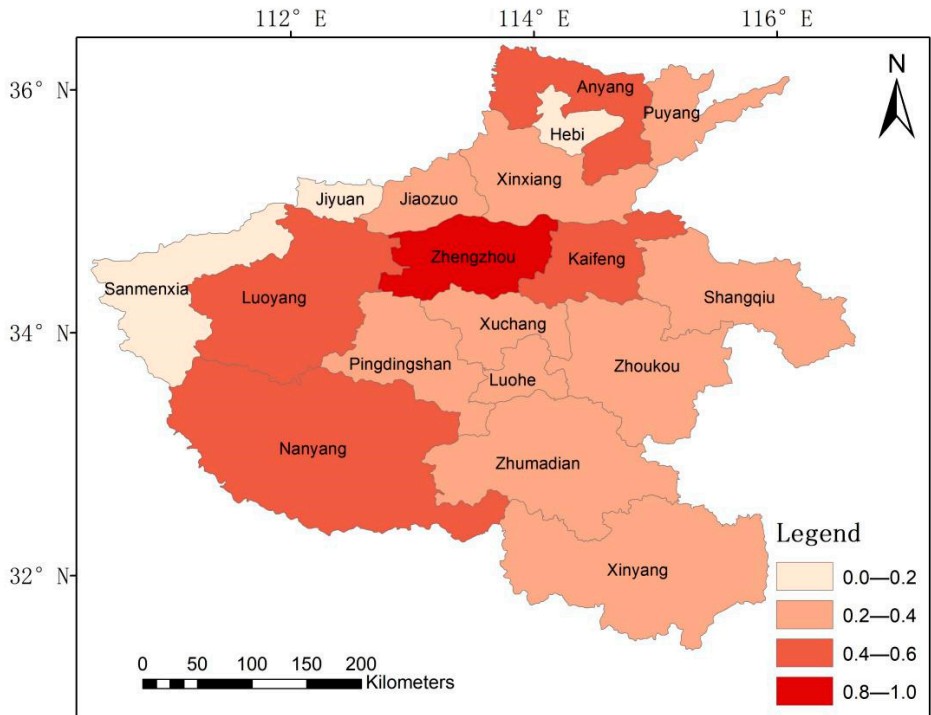

**Figure 11.** The results of Henan Province city grade classification.

**Table 5.** The results of this paper are compared with those of China.

| City | 2017 | 2018 | Q | Grade |
|---|---|---|---|---|
| **Zhengzhou** | 1 | 1 | 0.98 | 1 |
| **Anyang** | 4 | 4 | 0.57 | 3 |
| **Nanyang** | 3 | 3 | 0.53 | 3 |
| **Luoyang** | 3 | 3 | 0.50 | 3 |
| **Kaifeng** | 4 | 4 | 0.41 | 3 |
| **Zhoukou** | 4 | 4 | 0.39 | 4 |
| **Shangqiu** | 4 | 3 | 0.36 | 4 |
| **Xinxiang** | 4 | 3 | 0.33 | 4 |
| **Pingdingshan** | 4 | 4 | 0.32 | 4 |
| **Xuchang** | 4 | 3 | 0.32 | 4 |
| **Zhumadian** | 4 | 3 | 0.31 | 4 |
| **Puyang** | 5 | 4 | 0.26 | 4 |
| **Luohe** | 5 | 4 | 0.25 | 4 |
| **Jiaozuo** | 4 | 4 | 0.24 | 4 |
| **Xinyang** | 4 | 3 | 0.20 | 4 |
| **Hebi** | 5 | 5 | 0.19 | 5 |
| **Sanmenxia** | 5 | 5 | 0.15 | 5 |
| **Jiyuan** | 5 | 5 | 0.10 | 5 |

## 5. Discussion

Zhengzhou, the provincial capital, has always been the pacesetter of Henan Province and is far ahead of other cities in terms of comprehensive strength. In 2018, the GDP of Zhengzhou reached 1014.33 billion yuan, up 8.1% year on year, and the total economic output of Zhengzhou exceeded 1 trillion yuan. The economy of Zhengzhou accounts for 21.1% of the total economy of Henan Province. Therefore, the comprehensive index of Zhengzhou is better than that of other cities. Anyang and Luoyang are both ancient capitals with good economic development. Nanyang city has a large area, a large population, and a large economic aggregate. In addition to zhengzhou, these three cities also have a strong comprehensive strength.

Kaifeng is located to the east of Zhengzhou city. To promote economic development, the Henan provincial government put forward the policy of Zheng-Kai integrated development. This policy greatly promoted the development of the comprehensive strength of Kaifeng. The comprehensive strengths of Zhoukou, Shangqiu, Xinxiang, Xuchang, Pingdingshan and Zhumadian cities are relatively close.

The comprehensive strengths of Puyang, Jiaozuo, Luohe and Xinyang cities are also relatively close. Among them, although Xinyang city has a larger jurisdiction area, the development area of the city center is small, which is the reason that it reflects a weak comprehensive strength. Luohe and Xinyang are located on the most important railway line in China, the Beijing–Guangzhou railway, which has obvious traffic advantages and great development potential.

The comprehensive strengths of Hebi, Jiyuan and Sanmenxia are relatively weak. Among them, Jiyuan city is the only municipality directly under the central government of Henan Province that has good development potential. Although Hebi city has a small administrative area, it is located on the Beijing–Guangzhou railway route and has great development potential.

This paper proposes a method of city center detection and classification based on connectivity analysis. A connected operator was adopted to identify and divide the sub-center of the city and to determine the topological relationship between the adjacent urban centers, and different cities can be analyzed "crosswise" and "lengthwise" to generate a series of parametric information by constructing a tree structure, based on which, cities are classified and analyzed. In the process of city classification, compared with the national classification, this paper lacks the analysis of cultural and educational factors.

## 6. Conclusions

City classification can provide powerful data and technical support for city planning and government decision-making. In this paper, 18 cities in Henan Province were classified based on NTL images through connectivity analysis in mathematical morphology. The connectivity analysis of the NTL image can be carried out at different levels, and the tree-like structure can be constructed for "retrieving crosswise" and "retrieving lengthwise". The "crosswise" analysis mainly analyzes the connected region at the first level and generates certain indicators to evaluate and analyze the city. The "lengthwise" analysis mainly analyzes the tree structure generated in the urban central development area.

The "crosswise" analysis can mainly generate indicators of strength and area, while the "lengthwise" analysis can obtain the hierarchical depth and width of the tree structure of urban center development areas. Through this way of analysis, the city can be divided into two development models: (1) Multi-edge: the multi-edge development mode has a multilevel structure and few nodes in the urban center tree (e.g., Kaifeng and Shangqiu cities) and (2) Multicenter: the multicenter development mode has a structure with few levels and many nodes (e.g., Luoyang city). According to some indexes, cities in Henan Province can be classified and analyzed. The results show that the comprehensive index of Zhengzhou city, the provincial capital, is far ahead of that of other cities. In addition to Zhengzhou city, Luoyang, Nanyang and Anyang cities have good comprehensive indicators. The results of city classification in this paper are basically consistent with the 2017 and 2018 results of China's national classification, and only the results of Anyang and Kaifeng cities are different.

**Author Contributions:** Conceptualization, Z.Z., G.C. and C.W.; methodology, Z.Z., G.C. and C.W.; software, Z.Z.; validation, Z.Z., G.C. and C.W.; formal analysis, Z.Z., G.C., C.W., S.W. and H.W.; investigation, Z.Z.; resources, Z.Z.; data curation, Z.Z.; writing—original draft preparation, Z.Z.; writing—review and editing, Z.Z., G.C., C.W., S.W. and H.W.; visualization, Z.Z.; supervision, Z.Z.; project administration, Z.Z.; funding acquisition, Z.Z., G.C. and C.W. All authors have read and agreed to the published version of the manuscript.

**Funding:** This work was supported in part by the Funding for Post-doctoral Research Projects in Henan Province under Grant 1901018, and in part by the Doctoral Fund of Henan Polytechnic University under Grant B2018-24.

**Conflicts of Interest:** The authors declare no conflict of interest.

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
