# Peer review of "City Grade Classification Based on Connectivity Analysis by Luojia I Night-Time Light Images in Henan Province, China"

_remotesensing, doi:10.3390/rs12111705_

Round 1

Reviewer 1 Report

The manuscript aims to classify cities using a new sensor of nighttime data called “Luojia I Nighttime Light Image”.

1) The “Introduction” section of the manuscript requires extensive revision.

  1. A) The authors need to expand the review of literature that is relevant to their research. For example, the previous works that have been done in terms of city classification (the satellite data that have been used, attributes, etc.).

  1. B) The different applications of Luojia I Nighttime Light Image should be mentioned because it is a new sensor.

2) Page 2, lines 62-63. Give some examples of the problems.

3) Page 2, line 70. You mentioned that “In early 2013, NOAA/NGDC released a new generation of NTL data from the day/night band (DNB) of the Visible Infrared Imaging Radiometer Suite (VIIRS) on the Suomi National Polar-orbiting Partnership (NPP) satellite”. It is in 2011 not 2013 and the monthly data set is available from 2012.

4) Page 2, lines 74-75. You mentioned that “However, due to the low spatial resolution of DMSP-OLS and NPP-VIIRS NTL composite data, their applications are limited.” The application of the DMSP-OLS and NPP-VIIRS are not limited. The low spatial resolution of these data is a source of uncertainty. Thus, change the sentence especially the last part “their applications are limited”.

5) Page 3, line 96. Put a reference at the end of this sentence “Henan has a total area of 167,000 square kilometers, and its population is more than 96 million (the third most populous province in China)”.

6) Page 3, line 100. Put a reference at the end of this sentence “Henan, which has been regarded as "the center of the China" since ancient times, is located in the Jingjintang, Yangtze River delta, Pearl River delta and Chengdu-Chongqing urban belt”.

7) Page 3, lines 107-112. Put at least one or two references.

8) Page 4, Figure 2. Change the range of the scale bar to be such as (0     500      1000      1500     2000) for China and (0     50     100      150     200) for Henan province. Do this for all figures.

9) Page 5, lines 140-142. The equation 2 needs to be clarified.

10) Page 7, lines 187-192. Why you compute these attributes, Why they are important, Who used them?

11) Page 7, line 189. You have to put reference No, 18 at the end of this sentence “For “retrieving crosswise”, three statistics of NTL values were calculated at level 1 for each urban area, including maximum intensity (MAXIN), total intensity (TIN) and intensity standard deviation (INSTD)”.

12) Page 8, line 225, Results. It will be a good idea if the authors support their findings with other researches especially for Nanyang and and Zhengzhou cities.

13) Page 9, lines 231-233. You mentioned that “As shown in Table 3, the connected component attribute information of 18 cities is counted at level 1, and the maximum values per column are shown in bold font”. Nothing in bold font in Table 1.

14) Page 9, line 233. Chang “From the data in Table 3” TO “From the results in Table 3”.

15) Page 9, line 233. “From the data in Table 3, we can see that Nanyang city has the largest intensity value, which is related to the population size”. Write the intensity value (4.45) and the population size as well.

16) Page 9, line 244, Figure 8. Add two y-axes, one for the intensity value and the other for the population size.

17) Page 12, line 288. Add “respectively” at the end of this sentence “The LN and MNN of Pingdingshan city are 6 and 5”.

18) Page 13, line 299, Figure 10. Remove the background make it white.

19) Page 13, lines 321. “replace” is not the suitable word, change it.

20) Page 13, lines 323-325. No need for this paragraph, delete it.

21) Page 15, lines 352-353. You mentioned that “These three cities are cities with strong comprehensive strength except Zhengzhou”. I believe a word “except” not the suitable word.

Reviewer 2 Report

Dear authors,

although article is interesting, I believe that it can be improved.

Introduction does not provide sufficient background.

Figure 1 should be in chapter 2.

Figure 2 - areas are not clear to read - some of them overlap with border lines.

Figure 2 - for Scale SI units should be used (according to Instruction for authors)

Lines from 100 - 112 does not fit in this article. It is not referenced in any way. It does not show how this "facts" are important for your analysis.

Line 127 - Figure 3 first capital letter. Table 1 and Henan province also.

Figure 4 - just a suggestion - it could be useful to provide province boundaries in order to be more transparent.

Figure 5 should be on one page.

Lines from 202 to 205 missing reference.

Figure 7 - scale in SI units

Line 336 please avoid expressions like "were not exactly the same"

In discussion chapter you should include other authors work (that you will provide in introduction), compare your work with theirs and discuss it why your solution is better/worse in some cases.

Support Conclusions with results.

Reviewer 3 Report

The paper I have reviewed provide important insights to how remote sensing in particular night time
light, can help us to understand demographic changes in urban areas to update the city hierarchy.
The innovative combination of conventional nighttime light images to develop updated urban
systems is applicable for a wide range of policy makers. This complexity is well handled by the
authors.
The manuscript is well written and suitable for publication in the journal Remote Sensing
However, I have comments for the authors to address in order to strengthen the manuscript:
Comment 1:
In the introduction the authors are citing various research using within the theme city hierarchy
however they fail to mention Walther Christaller and his ground break findings. The manuscript
would benefit from including his findings.
Comment 2:
The present research is working on very fine spatial resolution but the part of the literature review
on Luojia I NTL images are not providing any insights to the limitations of the images and the
processing of the images.
Comment 3:
The discussion fails to relate the analysis conducted in the paper to other research findings. The
manuscripts need to relate the findings with previous research so the reader can be provided with
more accurate understadings of the strength and weaknesses of the current research.
Please address these comments and then the manuscript can be read again.

Round 2

Reviewer 1 Report

The manuscript has been improved and needs some edits.

1) Page 3, lines 96-99. Please add some of the following works as examples of Luojia I applications:

  • Article name “Mapping Urban Extent Using Luojia 1-01 Nighttime Light Imagery”.
  • Article name “Potentiality of Using Luojia 1-01 Nighttime Light Imagery to Investigate Artificial Light Pollution”.
  • Article name “Evaluation of Luojia 1-01 nighttime light imagery for impervious surface detection: A comparison with NPP-VIIRS nighttime light data”.
  • Article name “Improving population mapping using Luojia 1-01 nighttime light image and location-based social media data”.

2) Page 3, line 127. The numbers of the scale bare of China in Figure 1 is overlapped and unreadable. Let the scale bare starts from (0      500           1000             2000          3000). Moreover, remove the comma (Thousand's Separator) from the numbers. For example, 2000 instead of 2,000.

Put the north arrow in all figures.

3) Page 7, line 210. Delete the word “only”.

4) Page 9, line 264, Figure 7. Change the background of Figure 7 to be white and change the border lines of the cities to be black.

5) Page 15, Table 5. What is Q (third column) in Table 5.

Reviewer 2 Report

There are no further comments.

Author Response

There are no further comments.

Reviewer 3 Report

Ready to be published.

Author Response

There are no further comments.